# Structural basis of human Mediator recruitment by the phosphorylated transcription factor Elk-1

Didier Monté [1] ✉, Zoé Lens[1], Frédérique Dewitte[1], Marcus Fislage [2,3], Marc Aumercier[1], Alexis Verger [1] ✉ & Vincent Villeret [1] ✉

One function of Mediator complex subunit MED23 is to mediate transcriptional activation by the phosphorylated transcription factor Elk-1, in response to the Ras-MAPK signaling pathway. Using cryogenic electron microscopy, we solve a 3.0 Å structure of human MED23 complexed with the phosphorylated activation domain of Elk-1. Elk-1 binds to MED23 via a hydrophobic sequence PSIHFWSTLS[P]P containing one phosphorylated residue (S383[P]), which forms a tight turn around the central Phenylalanine. Binding of Elk-1 induces allosteric changes in MED23 that propagate to the opposite face of the subunit, resulting in the dynamic behavior of a 19-residue segment, which alters the molecular surface of MED23. We design a specific MED23 mutation (G382F) that disrupts Elk-1 binding and consequently impairs Elk-1-dependent serum-induced activation of target genes in the Ras-Raf-MEK-ERK signaling pathway. The structure provides molecular details and insights into a Mediator subunit-transcription factor interface.

The Mediator complex binds RNA polymerase II (Pol II), transmits regulatory signals from transcription factors to Pol II, facilitates assembly of the preinitiation complex (PIC), and stimulates phosphorylation of the Pol II C-terminal domain by cyclin-dependent kinase 7 (CDK7)[1–4]. The Pol II associated part of Mediator comprises 26 subunits in metazoans, organized into three modules[1,5]. The Head and Middle modules, together with the upper Tail, form a 21 subunits functional core that directly contacts the Pol II[6–10]. The remaining five subunits (MED15, MED16, MED23, MED24, and MED25) form the lower Tail module, which does not contact Pol II directly and whose main function is to connect Mediator to sequence-specific transcription factors[5,6,8,9,11–13]. Latest developments in the structural characterization of Mediator have reported structures of the 26 subunits Mediator, alone or in complex with Pol II and other cofactors forming the PIC[6–9]. These studies have improved our understanding of Mediator function by providing a more complete picture of the PIC complex. However, few structural details of transcription factor (TF) – Mediator interaction are available, owing to the disorder and dynamics of activation domains[14]. TFs control Mediator function, in part, by recruiting Mediators to specific genomic sequences, such as enhancers. The mechanisms underlying these processes are currently poorly characterized, and it is, therefore, important to make progress in elucidating the molecular mechanisms that determine TF-Mediator interactions.

The MED23 subunit is the largest subunit of the Tail module and is specific to metazoans. MED23 is the target of many transcription factors[15–19] and was originally discovered as a genetic suppressor of a hyperactive ras phenotype in *Caenorhabditis elegans*[20]. Studies in human and murine embryonic stem cells and fibroblasts have shown that one function of MED23 is to mediate transcriptional activation by the phosphorylated ternary complex transcription factor Elk-1 in response to the Ras-Raf-MEK-ERK (Mitogen-Activated Protein Kinase or MAPK) signaling pathway[16,21–27]. Elk-1 acts with the serum response transcription factor SRF to control many immediate-early genes such as Egr1 and Fos, and thereby controls the proliferation of several cell types. Knock-out of MED23 prevents activation by phosphorylated Elk-

[1]CNRS EMR 9002 Integrative Structural Biology, Inserm U 1167 – RID-AGE, Univ. Lille, CHU Lille, Institut Pasteur de Lille, Lille, France. [2]Structural Biology Brussels, Vrije Universiteit Brussel, Pleinlaan 2, Brussels, Belgium. [3]VIB-VUB Center for Structural Biology, VIB, Pleinlaan 2, Brussels, Belgium. ✉e-mail: Didier.Monte@univ-lille.fr; Alexis.Verger@univ-lille.fr; Vincent.Villeret@univ-lille.fr

1, without affecting many other transcription factors, underlying a direct link between MED23 and Elk-1[16,28]. MED23 knock-out leaves a functional Mediator complex[16,29], showing that this subunit can act as a specific target for certain transcription factors.

Activation of Elk-1 is regulated by phosphorylation in its transactivation domain (TAD, residues 310–428), located C-terminal to the DNA binding and SRF factor recruitment domains[26,30–32]. In addition to several S/T-P phosphorylation sequences, the TAD also contains two MAP kinase docking sites, the D-box, and the FQFP motif[32–35] (Fig. 1a). Alanine substitution of residues S383 and S389 abolishes activation in functional assays[22–24,36,37], indicating they are the most important functional phosphorylation sites, although other sites also impact activation. A kinetic study of the eight conserved phosphorylation sites in the Elk-1/SAP-1/NET family of ETS transcription factors showed that they exhibited distinct rates of phosphorylation and that this was regulated, in part, by the position of the sites relative to the ERK binding motifs[38]. In Elk-1, T368 and S383 were classified as fast sites, T353, T363, and S389 as intermediates sites, and T336, T417, and S422 as slow sites[38]. In addition to the crucial role of S383 and S389, progressive phosphorylation of other fast and intermediate sites further activated transcription, while late phosphorylation of slow sites slightly attenuated activation, thus providing a control of activity through progressive phosphorylation[38]. NMR analysis of unmodified and phosphorylated TAD Elk-1 showed that TAD populates largely disordered structural states, leaving open the question of its mode of interaction with MED23[38]. In addition to phosphorylation, other events must take place during the Elk-1/MED23 interaction, because the mutation of two hydrophobic residues in Elk-1 (F378 and W379) abolishes the recruitment of Mediator[22,24], without affecting the capacity of this mutated form of Elk-1 to be phosphorylated[24,39].

The structure of MED23 was previously determined by X-ray crystallography[40] at high resolution (2.8 Å), revealing that MED23 is structured (except for the last 30 amino acids), unlike many Mediator subunits (such as MED1 and MED15), which have large disordered regions[12]. More recently, the structure of MED23 was reported in the complete Mediator by cryo-EM and found to be consistent with the crystal structure[6,8,9]. Two forms of Mediators have been identified so far, called the Tail-extended (MED[E]) and Tail-bent (MED[B]) conformations, which differ in the relative orientation of the Tail module[8]. In both cases, the structure of MED23 is conserved and superimposable to the crystallographic isolated MED23 structure, except for the N-terminal 1–50 region (Supplementary Fig. 1), which is involved in contacts with subunits MED14 and MED15 in the Mediator complex. The structural integrity of MED23 in isolation is thus preserved and representative of MED23 in the full Mediator, validating its use in subsequent structural studies.

Here, we investigated the molecular basis for Mediator recruitment by phosphorylated Elk-1 through its interaction with Mediator subunit MED23. We report the high-resolution cryo-EM structure of MED23 in complex with the TAD of Elk-1, phosphorylated at three positions (T368, S383, and S389).

## Results

### Structure determination of free MED23 and Elk-1-TAD bound MED23 by cryo-EM

The crystal structure initially reported for MED23 required the use of a specific nanobody to crystallize the protein. The nanobody not only stabilized the structure through direct interactions with MED23 but was also heavily involved in crystal packing[40]. Crystallography did not prove to be an adequate technique for studies with a predicted disordered

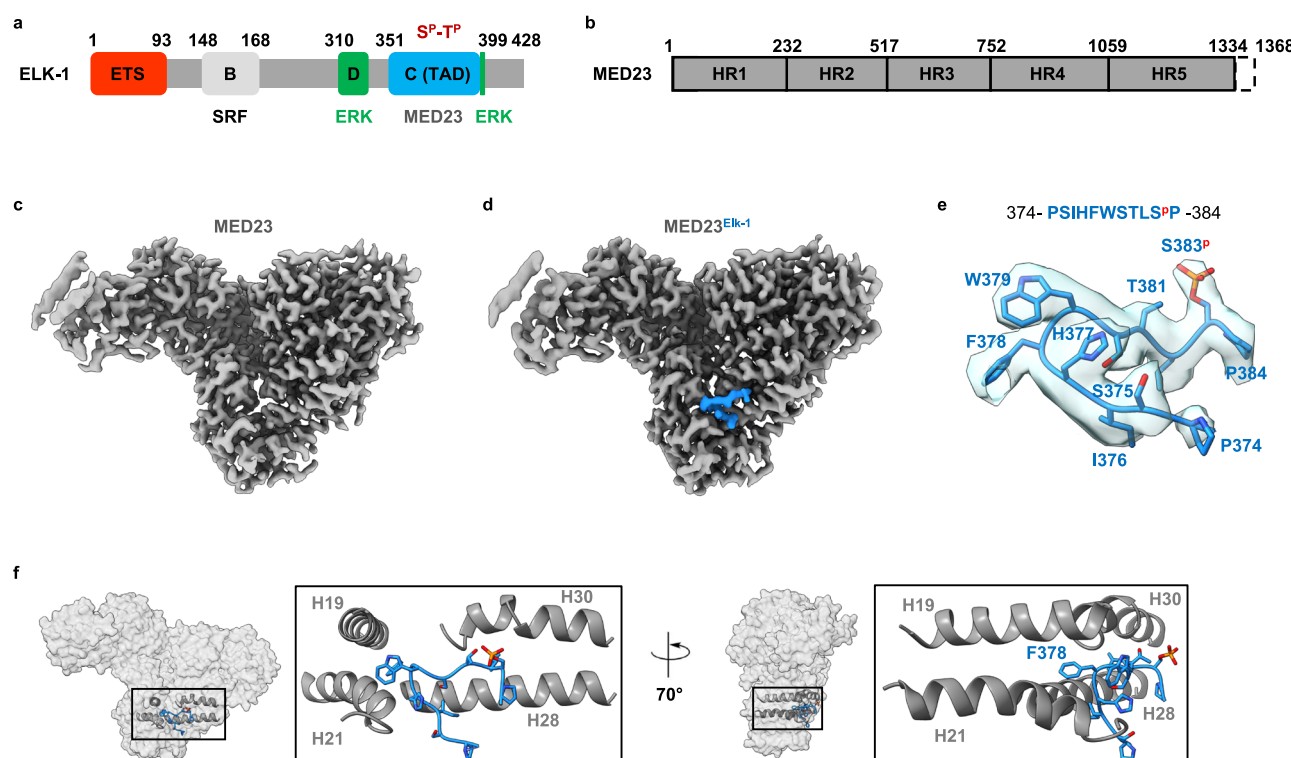

**Fig. 1 | Cryo-EM structures of MED23 and MED23[Elk-1] complex. a** Elk-1 domains organization. Five domains of Elk-1 are indicated. ETS: DNA binding domain (in red); B: B-box responsible for association with SRF (in light gray); D and FQFP: D-box and FQFP docking sites for ERK (in green) and C-domain: TAD (in blue) that contains eight conserved S/T-P phosphoacceptor motifs. **b** MED23 domains organization. Boundaries of HEAT-repeat regions (HR1 to HR5) of MED23 are indicated. MED23 is fully folded with the exception of the last 30 amino acids indicated by a black dashed box. **c** Cryo-EM map of MED23 and (**d**) Cryo-EM map of MED23[Elk-1] complex. **e** Map-model fit of Elk-1 (374–384) in surface representation at 2.6 sigma contour level. **f** Close-up view of MED23-Elk-1 (374–384) interface. In MED23, the Elk-1 binding site is formed by four helices: H19, H21 (from HR2) and H28, H30 (from HR3).

**Table 1 | Cryo-EM data collection, refinement, and validation statistics**

| | MED23 (EMD-50247) (PDB 9F76) | MED23^Elk-1 (EMD-50242) (PDB 9F6Y) |
|---|---|---|
| **Data collection and processing** | | |
| Magnification | 120000 | 60000 |
| Voltage (kV) | 300 | 300 |
| Electron exposure (e–/Å²) | 48 | 63.6 |
| Defocus range (µm) | − 0.8 to −1.7 | − 0.5 to −2.5 |
| Pixel size (Å) | 0.65 | 0.76 |
| Symmetry imposed | C1 | C1 |
| Initial particle images (no.) | 2.559.224 | 9.281.003 |
| Final particle images (no.) | 95.624 | 346.324 |
| Map resolution (Å) | 3.1/3.9 | 3.0/3.95 |
| FSC threshold | 0.143/0.5 | 0.143/0.5 |
| **Refinement** | | |
| Initial model used (PDB code) | 6H02 | 6H02 |
| Map sharpening B factor (Å²) | 124.3 | 137.5 |
| Model composition | | |
| Non-hydrogen atoms | 10179 | 10065 |
| Protein residues | 1258 | 1236 |
| R.m.s. deviations | | |
| Bond lengths (Å) | 0.002 | 0.003 |
| Bond angles (°) | 0.531 | 0.622 |
| Validation | | |
| MolProbity score | 1.40 | 1.73 |
| Clashscore | 5.77 | 8.51 |
| Poor rotamers (%) | 0 | 0 |
| Ramachandran plot | | |
| Favored (%) | 97.60 | 96.06 |
| Allowed (%) | 2.40 | 3.94 |
| Disallowed (%) | 0 | 0 |

partner, and we, therefore, considered using cryo-EM to study the interaction of MED23 with Elk-1. To validate the approach, we first determined the structure of MED23 alone by cryo-EM (Fig. 1b, c). We obtained a high-resolution structure of MED23, in which most of the structure is defined between 2.6-3 Å resolution, except for the N-terminal region, which appears more dynamic (Supplementary Figs. 2, 3 and Table 1). The structure is composed of α-helices that engage in HEAT-repeat motifs, forming five solenoid regions which are designated as HR1 to HR5 (Fig. 1b). MED23 core is composed of HR2 to HR5 regions and adopts a triangular shape in which HR2 to HR5 coil to position the C-terminal HR5 region in close proximity to HR1, which protrudes out of MED23 core (Supplementary Fig. 4). The overall shape of MED23 is that of an arch with a concave and a convex face. The structure is comparable to the previously reported crystal structure[40], with the main differences lying in the first N-terminal HEAT-repeat (residues 1–50), which appears dynamic (Supplementary Fig. 3b). In the crystal structure, this region is stabilized by nanobody-mediated crystal packing.

Given that transcriptional activation by Elk-1 through MED23 binding required phosphorylation[16,22,24,26,28,30,37–39], we reasoned that the best approach to characterize this interaction was to use the transactivation domain of Elk-1, including both binding sites for the ERK kinase. We, therefore, designed a construct (Elk-1^3P (308–401)) in which the TAD of Elk-1 was C-terminally fused to a GFP carrier protein. Elk-1^3P was co-expressed with ERK kinase, purified and phosphorylation status was assessed with Phos-Tag gel (Supplementary Fig. 5). We focused on the phosphorylation sites S383 and S389, which were shown to be important

for activation[36,39]. We also included T368, which was previously reported, as well as S383, to be a fast phosphorylation site and which, together with S383, surrounds the FW motif important for MED23 binding[38]. For all other phosphorylation sites, the serine and threonine residues were replaced by alanine residues. We validated the capacity of such a TAD (including 3 phosphorylation sites) to activate transcription (Supplementary Fig. 5). Before being deposited on cryo-EM grids, the MED23 and Elk-1^3P complex was incubated overnight, with a threefold excess of Elk-1^3P. We obtained a high-resolution cryo-EM structure of MED23 bound to Elk-1 (MED23^Elk1) (Fig. 1d), in which most of the structure is defined between 2.6-2.8 Å resolution, except for the N-terminal region, which appears more flexible, as observed for MED23 (Supplementary Figs. 6,7 and Table 1). We are thus in a favorable situation where the MED23 and MED23^Elk-1 structures are at comparable resolution.

### Elk-1 binds to MED23 through an eleven amino-acid phosphorylated sequence, PSIHFWSTLS^PP

Analysis of the MED23^Elk-1 electron density map reveals that Elk-1 binds to MED23 through a hydrophobic sequence PSIHFWSTLS^PP (hereafter referred to as «MED23 Binding Motif», or MBM) (Fig. 1d, e). The binding of MBM is observed on the concave face of MED23, within the MED23 core, at the interface between HR2 and HR3 regions (Supplementary Fig. 7d). In MED23, the MBM binding site is formed by four helices: H19, H21 (from HR2) and H28, H30 (from HR3) (Fig. 1f). Three residues from MBM, I376-Elk-1, F378-Elk-1, and L382-Elk-1, form hydrophobic interactions with MED23 (Fig. 2). Within MBM, the HFWS residues form a tight turn in which F378-Elk-1 binds deeply into MED23, is buried and sheltered by the adjacent H377-Elk-1 and W379-Elk-1 (Figs. 1f, 2). F378-Elk-1 is surrounded in MED23 by side chains from residues I339, L343 (H19), F379, G382, S383 (H21), and V533 and M537 (H28) (Fig. 2). Regions upstream and downstream of the HFWS turn in MBM follow the general direction imposed by the turn, and thus exit MED23 in opposite directions. Upstream of the HFWS sequence, I376-Elk-1 fits in a hydrophobic pocket made by residues from MED23 HR3: H534, M537, S538, H541 (H28) and L579 (H30) (Fig. 2). Downstream of the HFWS sequence, L382-Elk-1 is positioned in front of HR3 and fits in a hydrophobic pocket defined by residues H541, A544, I548 (H28), and L579, Q587, L588 (H30) (Fig. 2). L382-Elk-1 is adjacent to S383-Elk-1, which is phosphorylated. There is no extensive interaction between MED23 and this phosphorylated side chain, as it points outwards of the structure. No additional density coming from Elk-1^3P is observed outside of the MBM peptide (Fig. 1e). During the cryo-EM data analysis, we performed multiple rounds of 3D-classifications on the particles set corresponding to MED23^Elk-1, but never observed any additional electron density or other conformations. We also did not observe free MED23 particles, suggesting that the excess of Elk-1^3P used to prepare the cryo-EM grid was sufficient to shift the equilibrium toward the bound form. These observations support the conclusion that other regions of the TAD domain of Elk-1 remain highly dynamic upon binding to MED23.

Mutation of the central FW motif in the MBM peptide has a drastic effect and abolishes the interaction between Elk-1 and MED23 (Supplementary Fig. 8a), suggesting that these hydrophobic residues constitute much of the driving force for MED23 binding, in agreement with previous studies[22,38]. As Elk-1^3P includes three phosphorylation sites (T368, S383, and S389), we investigated their relative importance in MED23 binding using the SPR technique (Supplementary Figs. 8b, c). We first tested whether Elk-1 is able to bind to MED23 when phosphorylated only at S383 (Elk-1^S383P). We observed that Elk-1^S383P binds to MED23 with a $K_d$ of 81 nM (Supplementary Fig. 8c), whereas only minimal interaction was detected by pull-down assays in the absence of phosphorylation (data not shown), precluding any further binding estimation. This is consistent with the fact that phosphorylation at least at one position in Elk-1 is required for transcriptional activation. Comparable $K_d$ values of 42 nM and 60 nM were obtained for the doubly (Elk-1^{T368P-S383P}) and for the triply (Elk-1^3P) phosphorylated forms

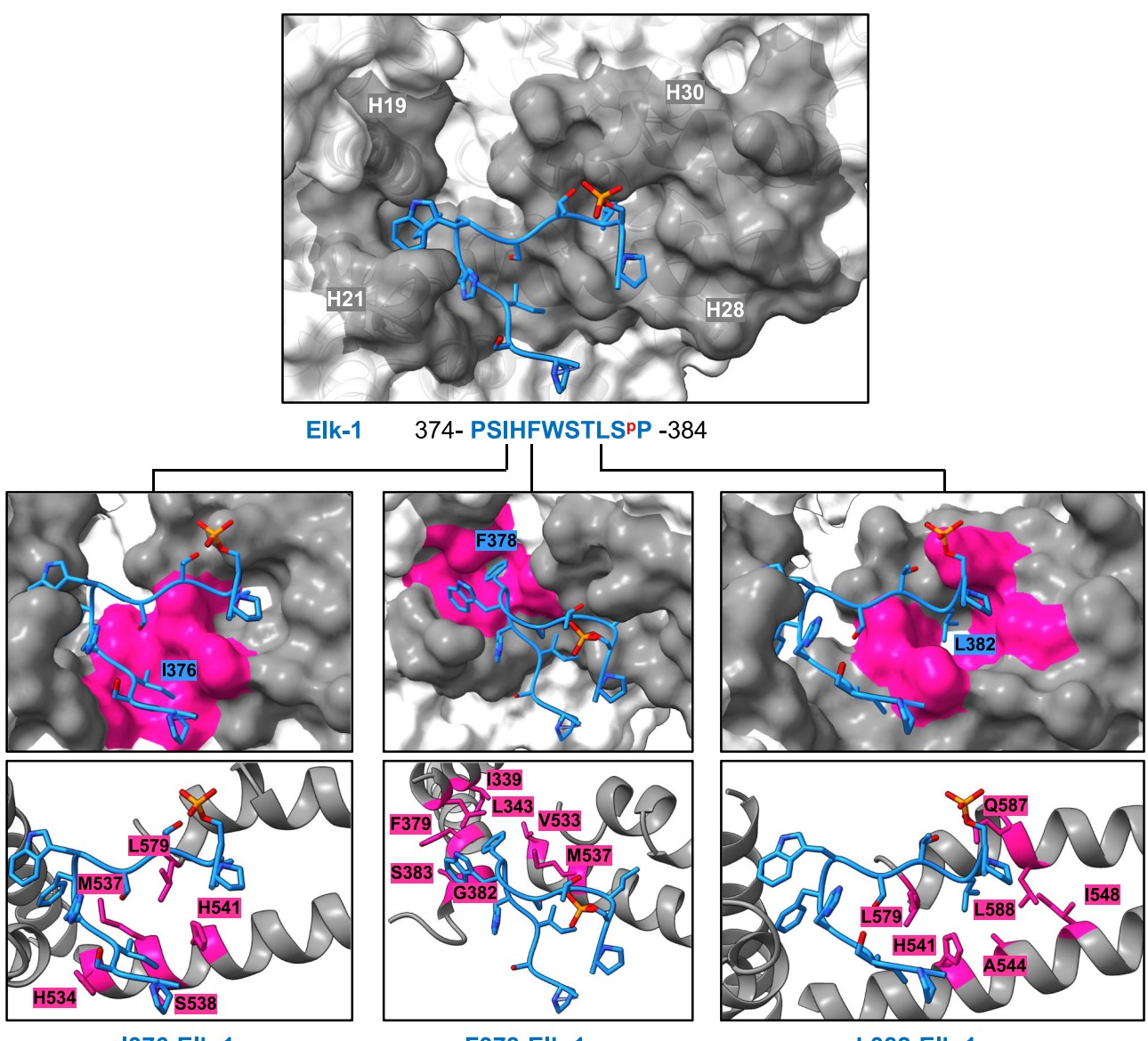

**Fig. 2 | MED23-Elk-1 interaction.** MED23-Elk-1 (374–384) interface related to Fig. 1f represented as a transparent surface (MED23 in white and MED23 Elk-1 binding site (H19, H21, H28 and H30 helices) in dark gray). Close-up views of I376-, F378- and L382-Elk-1 hydrophobic interactions with MED23 in surface (upper) and cartoon (lower) representation. Side chains of MED23 interacting residues within Elk-1 are shown in stick representation and colored in deep pink.

of Elk-1, respectively, suggesting that additional phosphorylation had no further impact on Elk-1 binding. We also tested whether phosphorylations at T368 and S389 were able to stimulate Elk-1 binding in the absence of phosphorylation at S383. We measured a $K_d$ of 75 nM for Elk-1$^{T368P\text{-}S389P}$, which is comparable to the $K_d$ measured for Elk-1$^{S383P}$. This suggests that phosphorylation at S383 in the MBM peptide is not strictly required for MED23 binding, but can be compensated, in our synthetic construct, by adjacent phosphorylations. It has long been suggested that phosphorylation of Elk-1 induces conformational changes within the transcription factor that are necessary for Mediator recruitment and further transcriptional activation[30,41]. Our results support this model and highlight the role of different phosphorylation events, showing that only one occurs within the Elk-1 peptide that directly interacts with MED23.

## Elk-1 binding promotes structural changes within MED23

Overall the core of MED23 and MED23$^{Elk\text{-}1}$ superimpose well for all HR regions (with mean rms deviations on Cα < ~ 0.6), except for the HR2

region, which appears slightly displaced upon binding of Elk-1 (Fig. 3a). A detailed comparison of MED23$^{Elk\text{-}1}$ with MED23 reveals that binding of Elk-1 to the concave face of MED23 promotes numerous localized structural changes around the MBM peptide, which propagate to the convex face of MED23, modifying its molecular surface (Fig. 3a–d). These molecular changes arise from a global movement of helix H19 in HR2 (Fig. 3c), necessary for MED23 to accommodate F378-Elk-1. In particular, I339 and L343 of MED23 move apart to free up enough volume for F378-Elk-1 binding, while F379 reorients to complement the hydrophobic environment around F378-Elk-1 (Supplementary Fig. 9). The relative movement observed for HR2 as a whole likely arises from these changes. Therefore, both helices H19 and H21 in HR2 are displaced (Supplementary Fig. 9b), while helices H28 and H30 from HR3 are unaffected upon Elk-1 binding. We also observed that the 501-516 segment located at the HR2-HR3 junction is disordered in MED23$^{Elk\text{-}1}$, whereas it is structured in MED23, where it folds into two short strands at the molecular surface (Fig. 3b). This dynamic behavior could result from HR2 movement relative to HR3 upon Elk-1 binding, although it is

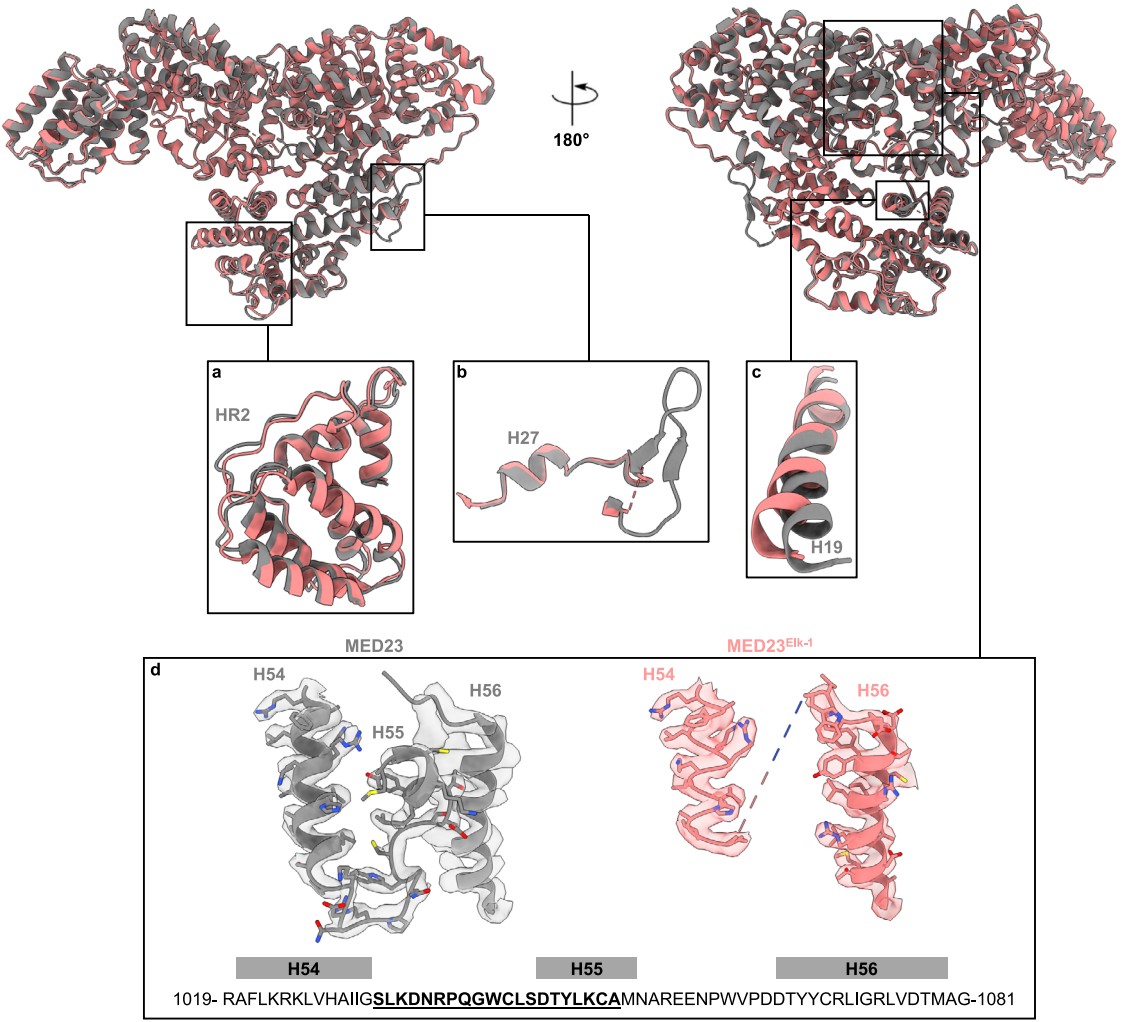

**Fig. 3 | Conformational changes in MED23 upon Elk-1 binding.** Overlay of MED23 (in dark gray) and MED23[Elk-1] (in light coral) structures in cartoon representation highlighting MED23 conformational changes upon Elk-1 binding. **a** Relative movement observed for MED23 HR2 domain residues 403–501. **b** MED23 501-516 segment located at the HR2-HR3 junction is disordered in MED23[Elk-1] complex. **c** Global movement of helix H19 in HR2. **d** Density differences reflecting HR4-lid conformational changes between MED23 and MED23[Elk-1]. The map-model fit of MED23 1019–1081 (in gray) and MED23[Elk-1] (in light coral) in surface representation. Side chains are represented in stick. Side chains of MED23 H56 were omitted for clarity. The 1019–1081 sequence is indicated (helices H54 to H56), with the HR4-lid sequence 1033–1052 underlined in bold.

unclear whether this modification has additional functional relevance. Helices H19 and H21 are almost perpendicular to H28 and H30 and connect the concave and convex faces of MED23 (Fig. 1f). Upon Elk-1 binding, the movement of H19 and H21 propagate toward the convex face, where further significant molecular rearrangements are observed. In particular, residues from S327 to S336 in helix H19 move up to 3 Å (Fig. 3c), interfering with the interaction network formed on the convex face at the interface of the HR2, HR4, and HR5 regions. Consequently, structural changes are observed, with a segment of 19 residues (S1033-A1052, hereafter named HR4-lid) which becomes dynamic or disordered, acting as a lid on the surface of MED23 (Fig. 3d).

The dynamics of the HR4-lid leads to a modification of the molecular surface of MED23, with the exposure of a new region - the HR4-lid dock site - at the junction of HR4 and HR5. The HR4-lid dock site is defined by structural motifs of HR4 (the H47-H49 segment, helices H52 and H53, and the H52-H53 segment) and HR5 (helices H56 and H58, and the H56-H58 segment). Although these motifs are generally conserved between MED23 and MED23[Elk-1], the H56-H58 segment appears slightly affected upon Elk-1 binding (Fig. 4). In MED23, this segment contacts HR4-lid and the HR2 region, in particular helix H19,

and is therefore ideally positioned to detect allosteric changes resulting from Elk-1 binding. Indeed, conformational changes are observed in the H56-H58 segment, which could be linked to the dynamic behavior of the HR4-lid. The side chain of W1092 reorients by 180° to contact residues L329 and H333 of H19 (Fig. 4b). This movement modifies the molecular network stabilizing the H56-H58 segment, which moves closer to the HR4-lid position and thus could induce its dynamic behavior and triggering. Another significant molecular event is the appearance of double conformations for H613 and H614, within HR3 (Supplementary Fig. 10). These histidine residues are in direct contact with the H52-H53 turn and the H56-H58 segment (Fig. 4b). Although the role of these double conformations is difficult to apprehend, they highlight the molecular network reorganization around the HR4-lid dock site and appear as a signature event upon Elk-1 binding, as they have not been observed previously (this work and unpublished results).

## Role of F378-Elk-1 for Elk-1 regulated transcription in a cellular context

Our structural study shows that an important event in the binding of Elk-1 to MED23 is played by F378-Elk-1, which will lead to the

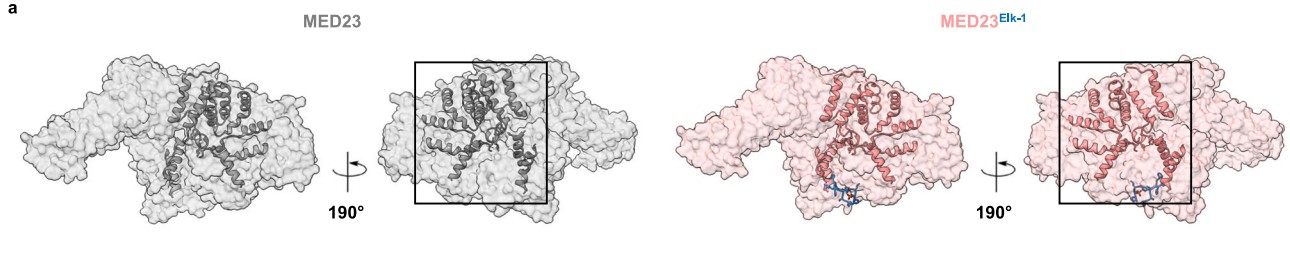

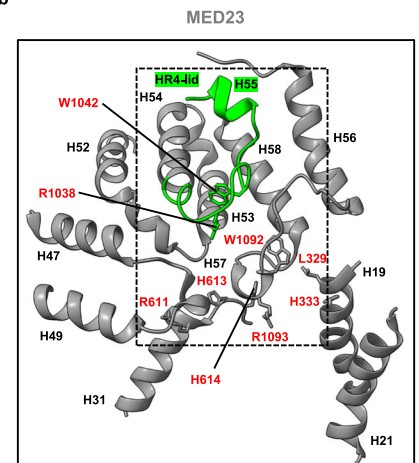 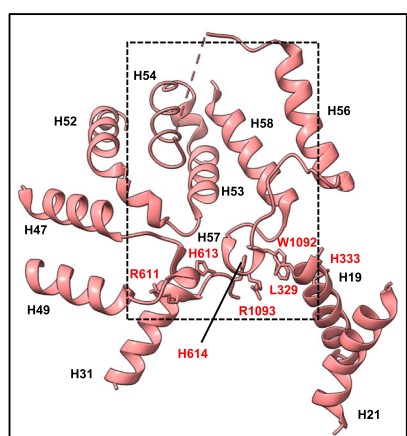 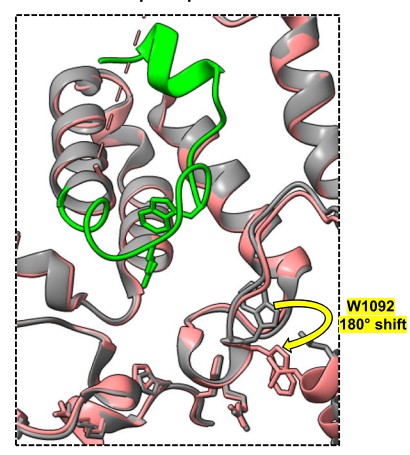

**Fig. 4 | HR4-lid structural motifs. a** Concave and convex views of MED23 and MED23^Elk-1 structures in transparent surface representation highlighting HR4-lid structural motifs in cartoon representation. **b** Close-up view of HR4-lid structural motifs. Elk-1 (374–384) was omitted for clarity. MED23 side chains discussed throughout the text are represented in stick and red color is used to label amino acid numbers. MED23 helices are shown as ribbons and black color is used to label the helices. The HR4-lid is colored and labeled in green. At right, the expanded view displays superimposed structures in ribbon form and shows residue W1092 that differed by 180° in side chain orientation between free- and bound-MED23.

recognition of the MBM peptide by MED23 as a turn and the subsequent propagation of allosteric conformational changes. The F378-Elk-1 binding site is bordered by G382, and the absence of a side chain in this position appears to be a prerequisite to free up enough space to accommodate F378-Elk-1 (Fig. 2). We observed that replacing G382 by a Phenylalanine (G382F MED23) would potentially obstruct the space occupied by F378-Elk-1, filling the cavity with a similar chemical group and preventing Elk-1 binding (Fig. 5a). WT and G382F mutant MED23 were expressed and purified from the baculovirus insect cell expression system and tested for their ability to bind phosphorylated GST-Elk-1^8P (308–428) in vitro. The recombinant G382F mutant failed to form a stable complex with Elk-1 compared to MED23 WT (Fig. 5b). This mutant was also evaluated in a cellular context, taking advantage of previously generated MED23^-/- MEFs[16]. We designed a luciferase reporter assay in which the GAL4 DNA binding domain is fused to the TAD domain of Elk-1 and tested whether luciferase activity could be observed in MED23^-/- MEFs where either MED23 WT or its mutated form G382F, had been reintroduced by transient transfection (Fig. 5c). As a control, we used the adenovirus E1A conserved region 3 (CR3), a potent activation domain that regulates early adenovirus genes and has been shown to bind MED23 to activate transcription[15,16,28]. Consistent with the hypothesis that the G382F mutation would block Elk-1 binding, the GAL4 reporter assay showed recovered expression upon reintroduction of MED23, unlike the G382F mutant, which showed no response (Fig. 5c). Interestingly, these results also showed that both MED23 WT and the G382F mutant were capable of supporting transcriptional activation by the E1A CR3 activation domain (Fig. 5c).

To investigate the functional effect of MED23 G382F mutant on the expression of the endogenous Elk-1 target gene Egr1, we generated from the MED23^-/- MEFs, cell lines stably expressing either the MED23 WT or the G382F mutant. Egr1 is one of the major target genes of Elk-1 in response to its activation by phosphorylation by the Ras MAPK signaling pathway[21,42,43]. We first assessed the transcriptional activity of GAL4-Elk-1^8P (308–428) and GAL4 E1A CR3 in these cell lines (Fig. 5d). In line with transient transfection (Fig. 5c), both WT and mutant MED23 cell lines supported E1A CR3 reporter gene expression, while GAL4-Elk-1^8P (308–428) TAD activity was completely abolished in MED23 G382F MEFs (Fig. 5d). These results highlight in a functional context our structural observations and also indicate that Elk-1 and E1A can bind to MED23 to different and non-overlapping sites.

Finally, we examined the ability of MED23^-/- MEFs stably re-expressing either the WT MED23 or the G382F mutant to initiate endogenous Egr1 expression after serum addition to serum-starved cells (Fig. 5e). Indeed, ectopic expression of MED23 WT rescued the serum induction of EGR1 while cells re-expressing the G382F MED23 showed reduced magnitude of serum induction of EGR1 comparable to MED23^-/- MEFs (Fig. 5e). *Egr1* as well as *Egr2, Egr3* and *Fos* mRNA levels were also analyzed by quantitative real-time PCR (qRT-PCR). We found that the mRNA level of these four Elk-1 target genes were higher in WT MED23 than in MED23 G382F MEFs (Fig. 5f). These results indicate that Elk-1 binding to MED23 via the MBM sequence is strictly necessary for transcriptional activation of Elk-1 target genes through the Mediator complex, validating in a cellular assay the structural observations reported here.

## Discussion

Although the human genome encodes more than 1600 TFs, only a fraction of these TFs has well-characterized molecular mechanisms[44].

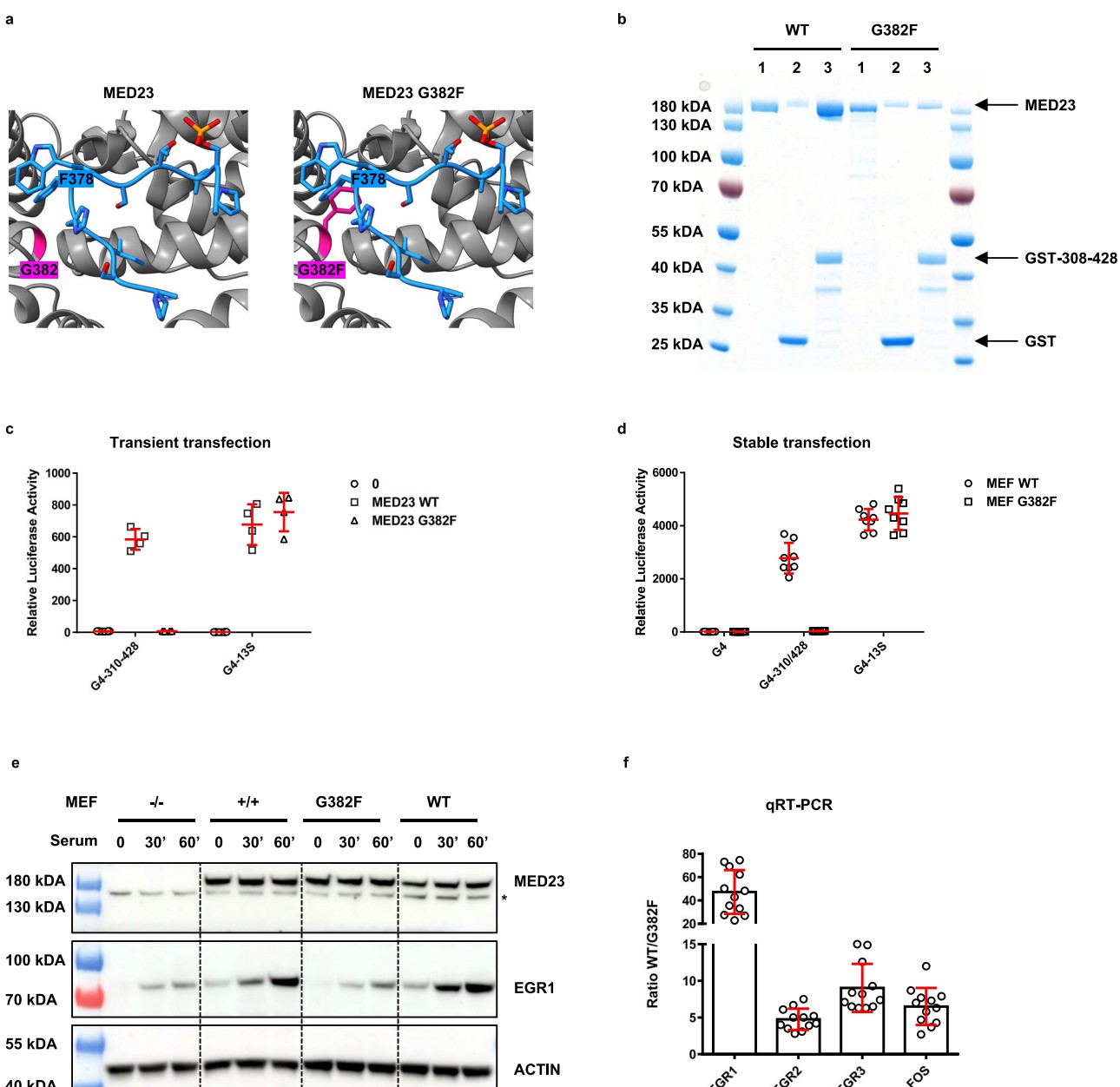

**Fig. 5 | MED23 G382F mutation impaired transcriptional activity of Elk-1.**
**a** Close-up view of MED23-Elk-1 (374−384) interface with MED23 G382 residue colored in deep pink. G382 was mutated in Phenylalanine using the ChimeraX rotamers tool, potentially obstructing the space occupied by F378-Elk-1. **b** Elk-1 TAD binding is disrupted by MED23 G382F mutation. GST and GST Elk-1$^{SP}$ (308−428) were immobilized on GST-Trap agarose and incubated with recombinant MED23 or MED23 G382F. Bound proteins were eluted with Laemmli sample buffer, resolved by SDS-PAGE, and visualized by Coomassie Blue staining. The molecular weight marker (in kDa) is indicated on the left. 1: MED23 WT and G382F input 1/40; 2: GST control and 3: phosphorylated GST Elk-1$^{SP}$ (308−428). **c** Defective Elk-1 TAD activity in MED23$^{-/-}$ MEFs. MED23$^{-/-}$ MEFs were transfected with a 5 × Gal4-E1B-TATA-luciferase reporter construct and a plasmid encoding either Gal4-Elk-1$^{SP}$ (308−428) or Gal4-E1A activation domain (13S CR3) with empty (0, circle), MED23 WT (square) and MED23 G382F (triangle) expression plasmids as indicated. Firefly luciferase activity was normalized to Renilla luciferase activity. Data are representative of three independent experiments and are presented as mean values ± SD. **d** MED23$^{-/-}$

MEFs stably expressing MED23 WT (MEF WT, circle) or MED23 G382F (MEF G382F, square) were generated and transfected as in Fig. 5c. DNA binding domain of Gal4 (G4) was used as control. Data are presented as mean values of two biological replicates ± SD. **e** Interaction between Elk-1 and MED23 is necessary in serum induction of Egr1. Time course of EGR1 protein expression after serum induction in MED23$^{-/-}$ MEFs, MEFs$^{+/+}$, and MED23$^{-/-}$ MEFs stably expressing MED23 WT or MED23 G382F. Cell lysates at the indicated minutes after serum addition to serum-starved cells were analyzed by Western blot with anti-MED23, anti-EGR1, and anti-Actin antibodies as indicated. The star indicates an unknown band. A molecular weight marker (in kDa) is indicated. **f** qRT-PCR analysis of immediate early genes transcription in MED23$^{-/-}$ MEFs stably expressing MED23 WT or MED23 G382F. The expression was normalized to GAPDH mRNA expression. Data represent fold change of *Egr1, Egr2, Egr3,* and *Fos* in MED23 WT relative to MED23 G382F MEFs. Data are presented as mean values (*n* = 4) of three biological replicates ± SD. Source Data are provided as a Source Data file.

Many contain disordered regulatory regions, which are further activated or inhibited by post-translational modifications such as phosphorylation[45,46]. Currently, more than 70 TFs have been reported to bind to various Mediator subunits, which belong to the three modules, Head, Middle, and Tail[8]. Depending on the targeted subunit and its location in the Mediator, it is very likely that each TF exerts certain specific modes of action, either for Mediator recruitment or for other Mediator functions, for example, in chromatin remodeling. However, few structural details of TF-Mediator interactions are available, owing to the disorder and dynamics of TF activation domains. To date, no high-resolution crystallography or cryo-EM data have been obtained for a TF-Mediator interface, either because such interactions are too dynamic or because they are technically difficult to trap by high-resolution techniques. Available evidence only shows that some activation domains that target Mediator MED15 or MED25 subunits retain their disorder even upon Mediator binding, adopting a so-called fuzzy interface[47–51]. These data were obtained by NMR studies using individual domains of their respective subunits. How TF-Mediator interactions will impact Mediator subunits and, as a result, the full Mediator, remains a central question. It is expected that TF-Mediator binding coincides with conformational changes in the Mediator complex, and this will influence Mediator function[52–55].

It has been known for decades that multisite phosphorylation is a major mechanism for regulating protein function[56]. However, at present, limited structural data are available to enhance our understanding of this process at the molecular level. Here, we reveal, at high resolution, how phosphorylated Elk-1 recruits Mediator by binding to the MED23 subunit. Elk-1 binds to MED23 via a hydrophobic sequence, PSIHFWSTLS$^P$P (MBM sequence), containing a phosphorylated residue, S383$^P$. There is no extensive interaction between MED23 and this phosphorylated side chain, as it points outward from the structure. Three hydrophobic residues of the MBM peptide— I376-Elk-1, F378-Elk-1, and L382-Elk-1—are central to the interaction. The MBM forms a tight turn around residues HFWS, which bind to MED23 between the HR2 and HR3 regions. The turn formation imposes the directionality of the Elk-1 chain both upstream and downstream of the MBM. F378-Elk-1 binds deeply into MED23 and is the residue where the Elk-1 chain changes orientation as the turn forms. In vitro and *in cellulo* experiments using MED23$^{-/-}$ cells show that F378-Elk-1 plays a crucial role in driving the Elk-1-MED23 interaction.

Numerous phosphorylation sites in Elk-1 influence the activation process[38]. Previous studies have suggested that phosphorylation of Elk-1 induces conformational changes within the transactivation domain, which are necessary for MED23 binding and subsequent transcriptional activation[22,30,41]. We present molecular data supporting this model and emphasize the role of different phosphorylation events, showing that only one, at S383, is located directly within the Elk-1 peptide that interacts with MED23. In the synthetic construct of Elk-1 used in our studies, the effect of S383 phosphorylation can be compensated by phosphorylation at adjacent sites. In a cellular context, where S383 is always present, additional phosphorylation events may contribute to recruiting other protein partners or inducing conformational changes within the full-length Elk-1 transcription factor. In addition to its transactivation domain, Elk-1 contains an N-terminal ETS DNA-binding domain and a B-box region that interacts with the SRF transcription factor. All these domains are linked together rather than functioning in isolation, and functional studies have suggested that phosphorylation of the transactivation domain induces a conformational change in Elk-1, which accompanies the loss of inhibition of DNA binding by the B-box, thereby stimulating DNA binding and SRF recruitment[41]. Multi-phosphorylation within the Elk-1 TAD can, therefore, act at different levels of regulation in full-length Elk-1.

The binding of Elk-1 to MED23 via the MBM sequence induces allosteric changes, that propagate from the concave face of the protein to its convex face, resulting in the dynamic behavior of a 19-residue lid sequence (HR4-lid), which becomes disordered. Consequently, the molecular surface of MED23 is modified in response to Elk-1 binding. The closed HR4-lid region is not predicted to be a protein-protein interaction site in free MED23, whereas the region of MED23 exposed upon HR4-lid alteration has all the characteristics of a protein-protein interaction site. This likely favors the hypothesis that, upon binding to Elk-1, MED23 modifies its molecular surface in order to recruit another partner. Indeed, MED23 has been reported to play important roles in post-recruitment steps[21], although molecular data are lacking. It regulates alternative mRNA processing via direct interactions with splicing factors[57], promotes Pol II into transcription elongation via direct interaction with CDK9[42], or promotes histone modifications on active genes[58]. Studies with Elk-1 mutants showed that certain histone modifications such as H4K16ac, H3K27ac, H3K9acK14ac, or H3K4me3 required Elk-1 to be not only phosphorylated but also bound to MED23, therefore competent to activate transcription[24]. The molecular surface changes observed in MED23 upon phosphorylated Elk-1 binding could be events related to the post-recruitment roles played by MED23.

An important question is to assess the potential impact of Elk-1 binding within the full Mediator. Mediator has currently been reported in both Tail extended (MED$^E$) and Tail-bent (MED$^B$) conformations[8]. The molecular events that induce one or the other conformation are not currently characterized and will require additional study, but in both cases, the position of MED23 within the Mediator is identified. Structural comparison of MED$^E$ and MED$^B$ shows considerable conformational differences in the Tail module, but the Head and Middle modules remain unchanged. These differences include the separation of MED23 from MED16/MED24, a shift of MED23 toward the Head module, and a large displacement and rotation of MED23[8]. Both in MED$^E$ and MED$^B$, MED23 makes only few contacts with the Head and Middle modules, that are limited to its HR1 region. Thus, based on current knowledge, one might hypothesize that conformational changes in MED23 upon Elk-1 binding are likely to have an impact mainly in the organization and relative orientation of the Tail module. The situation may differ for other transcription factors, for example those directly contacting the Head or Middle module. Whether and how the Tail subunits will reorient within Mediator upon Elk-1 binding to MED23 will require additional study. In the PIC complex, the concave face of MED23, which accommodates Elk-1 binding is oriented towards the 5' upstream DNA, ideally suited to interact with transcription factors (Supplementary Fig. 11).

Finally, considering the important role of MED23 and Elk-1 in Ras-MAPK-related pathologies[59], the present study might provide a molecular basis for further investigations aiming at targeting the MED23-Elk-1 complex in a therapeutic perspective.

## Methods

### Plasmids

pFastBac MED23 has been previously described[40]. pFastBac MED23-G382F mutant was obtained by PCR. MED23 with an N-terminal BAP (Biotin acceptor peptide) tag was obtained by PCR and cloned into pFastBac Dual vector with the BirA biotin ligase. MED23 and MED23-G382F sequences were then cloned into the PIN510A and FC550A expression vectors (SBI) to obtain the PIN510A-1 MED23, PIN510A-1 MED23-G382F, FC550A-1-MED23 and FC550A-1-MED23-G382F plasmids. The phiC31 integrase expression plasmid (FC200PA-1) is from System Biosciences (SBI).

Elk-1$^{3P}$ (308–401) containing the T336A/T353A/T363A mutations was synthetized and cloned into the pGEX-4T1 vector by Genscript and named thereafter Elk-1$^{3P}$ (308–401)-6His. pGEX-4T1 Elk-1$^{1P}$ $^{(S383P)}$ (308–401)-6His, pGEX-4T1 Elk-1$^{2P}$ $^{(T368P/S383P)}$ (308–401)-6His and pGEX-4T1 Elk-1$^{2P}$ $^{(T368P/S389P)}$ (308–401)-6His were obtained using the same procedure. pGEX-6P1 Elk-1$^{8P}$ (308–428)-6His was obtained by PCR using pCMV-Elk-1 as a template[33]. pGEX-6P1 Elk-1$^{8PFW>AA}$ (308–428)-6His was obtained by PCR. pGEX-6P1 13S (121–223) was obtained by PCR

using pCDNA3-13S as a template[60]. Elk-1[3P] was also cloned into the pET-6His-GFPm plasmid to obtain pET-GFPm Elk-1[3P] (308–401)-6His. These sequences were also cloned into the pcDNA3-Gal4 plasmid to obtain pcDNA3-Gal4 Elk-1[3P] (308–401), pcDNA3-Gal4 Elk-1[8P] (308–428) and pcDNA3-Gal4 13S (121–223). The reporter plasmid (Gal4)5-E1B-Luc and the pCMV-Renilla have been previously described[60]. pGEX-4T3-HA-ERK2 GOF was a gift from Christopher Counter (Addgene plasmid # 53200) and pGEX-4T-1 3xHA-MEK1-DD was a gift from Kevin Janes (Addgene plasmid # 47576). ERK2-GOF and MEK1-DD sequences were cloned into pColaDUET (Novagen) to obtain the pColaDUET MBP-ERK2-GOF/MEK1DD co-expression vector. All plasmids are listed in the Supplementary Table 1.

## Protein expression and purification

MED23 and MED23 G382F were produced as described previously[40]. Briefly, DNA encoding human MED23 isoform 1 (NM_004830.3) was amplified and cloned into a pFastBac containing a rhinovirus 3 C protease site followed by a 6xHis tag. Recombinant baculovirus were obtained using the Bac-to-Bac expression system (Invitrogen). After baculovirus infection, ExpiSF9 cells (Thermo) were resuspended in a lysis buffer. Purification of MED23 was performed on a Talon Resin column (Clontech). Beads were washed with lysis buffer, and MED23-6His was eluted using the same buffer supplemented with 250 mM Imidazole. MED23 was purified by size-exclusion chromatography on a Superdex 200 column (GE Healthcare). The MED23-6His and MED23 G382F-6His proteins were then concentrated to 2 mg ml⁻¹, and aliquots were flash-frozen and stored at − 80 °C. BAP-MED23-6His was purified using the same procedure. GST and GFP fusion proteins were expressed in *E. coli* strain BL21 (DE3), grown at 37 °C in LB medium (2 L) to an optical density of 0.8 at 600 nm, and expression was induced with 0.5 mM IPTG overnight at 18 °C. The pColaDUET MBP-ERK2-GOF/ MEK1DD plasmid was co-transfected with the Elk-1 derivatives to allow in vivo phosphorylation. Recombinant proteins from soluble lysates were purified on GSTrap 4B and/or HisTap HP column (Cytiva) as previously described[60]. In vivo, phosphorylation of GFP-Elk-1[3P] (308–401)-6His and GST-Elk-1[8P] (308–428)-6His recombinant proteins was assessed by SuperSep Phos-tag 12.5% precast SDS-PAGE gel (Fujifilm). As no phosphorylation control, these proteins were treated with lambda-protein-phosphatase according to the manufacturer's protocol (NEB).

## Sample preparation for cryo-EM studies

MED23 was prepared at a concentration of 0.5 mg/ml in a buffer containing 20 mM Tris/HCl pH 7.5, 100 mM NaCl, 1 mM TCEP. For MED23, the protein was deposited on a glow-discharged Quantifoil R1.2/1.3 Cu 200 mesh grid and plunge-frozen in liquid ethane using a Vitrobot machine, with the following setup: temperature at 4 °C, humidity ~ 95%, blotting time 5.5 sec, blot force 2. For MED23[Elk-1], the complex was obtained by mixing MED23 with Elk-1[3P] at a molar ratio of 1:3, in the same buffer. The complex was incubated overnight and then deposited on a glow-discharged Quantifoil R2/1 Cu 300 mesh grid and plunge-frozen in liquid ethane, using a Vitrobot machine configured as with MED23.

## Cryo-EM data collection and processing

For MED23 (Supplementary Fig. 2), a total of 3799 movies were acquired on a Titan Krios G4 transmission electron microscope at the Thermo Fisher Scientific NanoPort facility, located in Eindhoven, the Netherlands. The movies were imaged at a nominal pixel resolution of 0.65 Å per pixel using a Falcon 4 detector. Each movie was collected across 40 frames with a total electron dose of 48 e⁻ Å⁻² (Supplementary Fig. 2 and Table 1). The movies were aligned with the software MotionCor2[61], and their contrast transfer function (CTF) parameters calculated by Ctffind4[62]. From these micrographs, 2.599.224 particles were picked using Topaz with the ResNet16 pretrained model[63], and

extracted at 4Xbinning. The particles were subjected to a single round of 2D classification to remove most junk/ice contaminations. Selected particles (548.597 particles) were further sorted into ten separate reference-free 3D classes using the ab initio module of CryoSparc[64]. Only one class, with 109.575 particles, corresponded to folded MED23 particles, the others representing either junk or degraded particles. This class was re-extracted with a binning factor allowing to reach a theoretical resolution of 2.03 Å and submitted to non-uniform refinement. Rounds of heterogeneous refinements allowed to remove particles that showed heterogeneities in the N-terminal region of MED23. This step yielded the final data set containing 95.624 particles, which was submitted to non-uniform refinement, resulting in an electron density map with a global resolution (reported as Fourier shell correlation of 0.143) of 3.1 Å. A resolution map with a 0.5 threshold for local FSC resolution estimation was calculated in CryoSparc and is showed in Supplementary Fig. 3. The core of MED23 is well resolved at resolutions below 3 Å (2.6 Å-3 Å), except for small regions in loops. Only the HR1 region is less well defined: residues 1–50, the segment which contains the three N-terminal helices, only showed blobs of density, and thus, they have not been included in the model. Starting from helix H4, HR1 gets better defined, and from helix H7, it displays a resolution around 3.5 Å or better. From helix H12, the resolution of the map is around or below 3 Å.

For MED23[Elk-1] (Supplementary Fig. 6), a total of 9924 movies were acquired on a JEOL CryoARM300 transmission electron microscope equipped with an omega energy filter at the BECM (Biological Electron Cryogenic Microscopy) VIB-VUB Facility in Brussels, Belgium. The movies were imaged at a nominal pixel resolution of 0.76 Å per pixel using a K3 direct electron detector in counting mode (Gatan). Each movie was collected across 60 frames with a total electron dose of 63.6 e⁻ Å⁻² (Supplementary Fig. 6 and Table 1). The movies were aligned with the software MotionCor2[61], and their contrast transfer function (CTF) parameters calculated using CryoSparc[64] ver.4.4.1. A total of 8420 micrographs remained after curating the data with a low-resolution cutoff of 6 Å for the CTF fit resolution. From these micrographs, 9.281.003 particles were picked using Topaz running with the ResNet16 pretrained model[63] and extracted at 4Xbinning. The particles were subjected to a single round of 2D classification to remove most junk/ice contaminations. The selected particles (2,897,659 particles) were then sorted into six separate reference-free 3D classes using CryoSparc's ab initio module, configured with a low expected class similarity factor. Only one class (903.091 particles) corresponded to MED23 particles, the remaining five classes representing junk or degraded particles. Another round of classification, using the ab initio module configured with a high expected class similarity factor, was carried out up to 6 Å resolution, to sort the particles stack into three classes. The main class (434.074 particles) corresponded to the most complete MED23 structure, while the two others displayed only the MED23 core, the N-terminal region (HR1 throughout the text) appearing disordered or degraded.

The particles from these three classes were re-extracted in larger boxes with a binning factor allowing to reach a theoretical resolution of 2.375 Å, cleaned to remove too close or duplicated particles, and submitted to non-uniform refinements. The particle stack corresponding to full MED23 (346.324 particles) refined to a global resolution, reported at Fourier shell correlation of 0.143, of 2.98 Å. Further 3D classifications did not allow the identify other conformations or any free MED23 particles. Various parameters controlling mask extent and threshold were evaluated, but always resulted in a similar electron density map. Further splitting of this set of particles only degraded the electron density map, as revealed by visual inspections. It is likely that this results from a degradation of the signal/noise ratio from a reduction in the redundancy of the orientations. A resolution map with a 0.5 threshold for local FSC resolution estimation was calculated in CryoSparc[64] and is showed in Supplementary Fig. 7. The distribution of

electron density definition is similar to that observed for MED23, although overall slightly better. Here, also the core of MED23 is resolved at resolutions below 3 Å (mostly 2.6 Å - 2.8 Å), except for small regions in loops. The HR1 region is less well defined, with residues 1–50 only showing blobs of density, and thus not included in the model. Starting from helix H4, helices from HR1 get better defined and, from residue ~P93 they display a resolution around 3.5 Å or better. From residue ~ G194, the resolution of the map is around or below 3 Å. The Elk-1 peptide is also defined at a resolution below 3 Å, except for the N and C-terminal prolines which appear more dynamic (Supplementary Fig. 7). Of note, the distribution of particle orientations differs between MED23 and MED23$^{Elk-1}$, which could be due to the addition of a protein with a specific mass and size in the MED23$^{Elk-1}$ study. This protein consists of the TAD domain of Elk-1 fused to a GFP tag. The added mass may influence the behavior of MED23 particles in solution and during the vitrification process.

The two classes that showed disordered or degraded HR1 after the second run of ab initio classification were also submitted to non-uniform refinements, and the respective electron density maps reached the global resolution of ~ 3.3 Å (Supplementary Fig. 6). Visual inspection revealed that in one case most of the HR1 region was missing from the electron density map, not even showing large density blobs, which might indicate altered or degraded protein. Nevertheless, the density corresponding to Elk-1 was also clearly visible, and well defined. In the second electron density map, the degradation in the N-terminal region was even more pronounced and also affected HR2 up to residue C297, in addition to HR1. The density corresponding to Elk-1 was still visible. These two classes were not used further (Supplementary Fig. 6).

### Model building, refinement and validation
For model building, the MED23 and MED23$^{Elk-1}$ maps were processed by deepEMhancer[65], in both cases using the experimental half-maps coming from the non-uniform refinements. The MED23 and MED23$^{Elk-1}$ models were built by docking the crystallographic MED23 structure (PDB entry code 6H02) into the cryo-EM maps. Docking was done using ChimeraX software[66]. The two models were inspected visually and manually corrected using Coot[67]. For the MED23$^{Elk-1}$ model, the region corresponding to Elk-1 was easily constructed. The models were then subjected to real-space refinement using the Phenix software[68]. A few rounds of refinement and visual inspection were applied to finalize the model construction. The definition of MED23 structural elements corresponds to the crystallographic structure[40]. Figures were prepared with ChimeraX[66]. The deepEMhancer processed maps are used in all illustrations.

### Cell culture and starvation experiments
MEFs were maintained in DMEM containing 10% FBS. For serum starvation experiments, $100 \times 10^3$ cells plated on 6-well plates were cultivated in 0.1% FBS. 48 h later, cells were stimulated with 10% FBS for 30 min or 1 h. Cells were then washed with cold PBS and lyzed in 100 μl RIPA buffer. Proteins were analyzed by western-blot using BD Pharmigen (anti-MED23 550429), Cell Signaling anti-Egr1 (15F7), and Invitrogen anti-beta-actin-HRP (BA3R) (Supplementary Table 1).

### Stable transfection
MED23$^{-/-}$ MEF cells were seeded overnight at $160 \times 10^3$ cells per well on 6-well plates and co-transfected with FC550A-1-MED23 or FC550A-1-MED23-G382F and the FC200PA-1 helper vector. FC200PA-1 transiently express the phiC31 integrase to insert the expression cassette into pseudo attP genomic sites using the attB sites in the FC550A-1 backbone according to the manufacturer's protocol (SBI). The cells were then selected with 2 μg/ml puromycin, and MED23 expression was verified by western blot (BD Pharmigen anti-MED23 550429). The resulting cells were respectively named MEF-WT and MEF G382F.

### Transient transfection and luciferase reporter assay
Cells were seeded overnight at $40 \times 10^3$ cells per well on 24-well plates and transfected by Lipofectamine 2000 (Invitrogen) with a luciferase Gal4 reporter plasmid 5xGal4-E1B (100 ng) and pCMV-Renilla (5 ng) along with various expression constructs as indicated (PIN510A-1, PIN510A-1 MED23 and PIN510A-1 MED23-G382F). 24 h post transfection, cells were lyzed and subjected to Dual-luciferase reporter assays according to the manufacturer's protocol (Promega).

### Real-Time PCR
qRT-PCR was performed as previously described[60]. Briefly, total RNA was isolated from MED23 WT and MED23 G382F MEFs using TRIZOL. The first-strand cDNA was generated, and real-time PCR was performed using a SYBR Green PCR master mix. Primer sequences used in the experiments were adapted from ref. 69.

### Pull-down assay
The interaction between MED23 and Elk-1 was measured in GST pull-down experiments. 4 μg of GST constructs were immobilized on 25 μl of GST-Trap (Chromotek) slurry for 1 hour in PBS-casein containing proteases and phosphatases inhibitors at room temperature. After PBS washing, beads were incubated with 20 μg of recombinant MED23 or MED23 G382F for 90 min at 4 °C in PBS-casein. After 3 washing steps, bound proteins were eluted, separated on a 4-20% precast SDS-PAGE gel (Genscript), and stained with ready to use Quick Coomassie stain solution (NeoBiotech) (Supplementary Table 1).

### Surface plasmon resonance
Affinity measurements were performed on a BIAcore T200 optical biosensor instrument (Cytiva). Recombinant MED23 with a BAP tag was biotinylated during expression. Capture of biotinylated MED23 was performed on a streptavidin SA sensor chip in PBS + buffer (Cytiva). One flow cell was used as a reference to evaluate unspecific binding and provide background correction. Biotinylated-MED23 was injected onto a SA chip (Cytiva) at a flow rate of 30 μL/min until the total amount of captured MED23 reached between 2000 and 3000 resonance units (RUs) according to MED23 batch-to-batch variation. Elk-1 derivatives fused to GST (Elk-1$^{3P}$, Elk-1$^{2P\,(T368P/S383P)}$, Elk-1$^{1P\,(S383P)}$ and Elk-1$^{2P\,(T368P/S389P)}$) were injected with increasing concentrations ranging between 0.25 μM and 4 μM in 5 half dilution points in a single cycle. Single-Cycle Kinetics (SCK) analysis was performed in triplicate to determine association $k_{on}$ and dissociation $k_{off}$ rate constants by curve fitting of the sensorgrams using the 1:1 Langmuir model of interaction of the BIAevaluation software 2.0 (Cytiva). Dissociation equilibrium constants ($K_d$) were calculated as $k_{on}/k_{off}$.

### Reporting summary
Further information on research design is available in the Nature Portfolio Reporting Summary linked to this article.

## Data availability
The cryo-EM maps for MED23 and MED23$^{Elk-1}$ have been deposited to the Electron Microscopy Data Bank (EMDB) under the accession codes EMD-50247 and EMD-50242. The corresponding atomic coordinates are deposited in the RCSB Protein Data Bank under the accession codes 9F76 and 9F6Y. Source data are provided in this paper.

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

## Acknowledgements

We thank Dr Arnold Berk (UCLA, Los Angeles, USA) for MED23[-/-] MEFs and MEFs[+/+]. We also thank Dr. Deniz Ugurlar for data acquisition at Thermo Fisher Scientific NanoPort facility in Eindhoven, the Netherlands, Dr. Davy Sinnaeve (EMR 9002 CNRS) for sharing computational resources acquired in the context of the ANR-20-CE29-0015 URANUS, Clément Danis and Elian Dupré for helpful discussion about SPR and Professor Andy Sharrocks for pCMV-Elk-1 plasmid. This work has been supported by an Emergence Grant from the Cancéropôle Nord-Ouest and by an ANR Grant MEDNET (ANR-16-CE12-0021). We would like to thank the VIB-VUB Facility for Bio Electron Cryogenic Microscopy (BECM) for support during Cryo-EM data collection. This paper is dedicated to the memory of our dear colleague Dr Bernard Clantin, who passed away in 2022.

## Author contributions

D.M., Z.L. and F.D. carried out cryo-EM sample preparations and cellular assays. Z.L. and V.V. prepared cryo-EM grids and performed grid characterizations. M.F., Z.L. and V.V. carried out data collection. V.V. performed data processing, map construction, and structure refinement. M.A., D.M., A.V. and V.V. analyzed the data. V.V. wrote the manuscript, with contributions from D.M., A.V., Z.L. V.V., D.M. and A.V. initiated the project and conceived experiments. All the authors contributed to the reviewing and editing of the manuscript.

## Competing interests

The authors declare no competing interests.
