## [Transparent Peer Review file · Nature Communications]

Structural basis of human Mediator recruitment by the phosphorylated transcription factor Elk-1

Corresponding Author: Dr Vincent Villeret

Version 0:

Reviewer comments:

Reviewer #1

(Remarks to the Author)

Mediator is a multi-subunit complex that connects transcription factors (TFs) with the RNA Polymerase II machinery to regulate transcription. Many TFs have been reported to bind to different Mediator subunits. However, the mechanisms of TF-Mediator interactions remain unclear due to the diversity and dynamic nature of transcription factors. The work by Didier Monté et al. describes the structure of the human Mediator subunit MED23 with the activation domain of the transcription factor Elk-1. The authors show that the Elk-1 activation domain binds MED23 through a hydrophobic region, inducing conformational changes in MED23 and revealing the interaction mechanism between the MED23 subunit and Elk-1. This work provides new insights into the TF-Mediator interaction mechanism and how TFs interact with Mediator to regulate gene transcription. I support the publication of the study.

Comments:

1. The multiple phosphorylation sites in Elk-1 regulate its transcriptional activation. The authors discussed that Elk-1 S383p is directly involved in MED23 binding and contributes to the stabilization of MED23-Elk-1 interaction. Would the lack of phosphorylation at S383 weaken the binding between MED23 and Elk-1?
2. The manuscript identifies the key residue, F378-Elk1, which plays a crucial role in the interaction between MED23 and Elk1. The authors demonstrate that the MED23-G382F mutation disrupts binding to Elk-1, thereby affecting the expression of the Elk-1 target gene Egr1. How does MED23-G382F affect the transcription levels of Elk-1 target genes in MED23-G382F mutant cells?
3. The authors should provide the SDS-PAGE of purified protein to show the quality of the proteins used in the study.

Reviewer #2

(Remarks to the Author)

Monte et al. determine a cryo-EM structure of the Elk1-binding mode of human MED23 at a reported resolution of 3.0 Å. They propose a model of the path of Elk1 to the binding site of MED23. As Mediator is one of the most critical co-activators for transcription, this paper potentially can capture attentions from general readers. However I feel there are several deficiencies. Most critically, it is unclear how the structural information gleaned from this study using Elk1 will specifically lead to new understandings of transcriptional mechanisms. The dissociation constant (Kd) of MED23 to this region of Elk1 should be defined. Furthermore, how much the HFWS sequence contributes to the binding in terms of Kd, as well as how phosphorylations contribute to the Elk1 binding in terms of Kd, should be elucidated. While the authors provided some information by citing previous papers, these citations are largely descriptive. Given that the authors seem to have a good expression system and pure soluble MED23, these measurements should be feasible without technical difficulties. If these binding parameters are not addressed, it is unlikely that this paper will capture the attention of general readers in Nature Communications.

Other comments:

Line 76-77: "Medium to low resolutions"

It's too vague to say this. Medium to low means resolutions lower than 5-6 Å.

Line 140: S383p-Elk-1 may form an internal hydrogen bond with the hydroxyl group of T381-Elk-1, as well as with Q587 from MED23.

Readers cannot judge the presence of hydrogen bonding based on Fig. 1.

Line 144: No additional electron density coming from...

Cryo-EM does not image "electron density" unlike X-ray crystallography. People just say density or cryo-EM density.

Line 153: Although they do not appear to be directly involved in MED23 binding, they enhance the activation process and thus could play a role in promoting the propensity of the MBM peptide to adopt a turn conformation capable of binding to MED23. In addition, they could promote the recruitment of other protein partners.

I think this is too speculative. Authors should provide info about how much T368p and S389p as well as S383p contribute to binding of the Elk1 peptide to MED23 in terms of Kd.

Line 324: In the PIC, the concave face of MED23, which accommodates Elk-1 binding, is oriented towards the 5' end of the promoter DNA, ideally suited to interact with potential upstream DNA sequences (Extended Data Fig. 10).

This statement is confusing. Is the Elk1 binding site immediately upstream of the core promoter? I thought it is several kbp away from the core promoter.

Table 1

The table should include map-model FSC (0.5).

Extended Data Fig. 2 and Fig. 6

Micrograph and 2D classes should have scale bars.

Extended Figure 5b

It would help to label the bands seen to convince the reader of the purity and phosphorylation status

Reviewer #3

(Remarks to the Author)

Monté et al., Structural basis of human Mediator recruitment by the phosphorylated transcription factor Elk-1

The authors of this study follow on their earlier determination of the first X-ray crystallography structure of Mediator subunit MED23 by now reporting cryo-EM maps (and corresponding atomic models) for MED23 and MED23 bound to the core portion of the trans-activation domain (TAD) of transcription factor Elk-1.

STRENGTHS

- The molecular analysis to identify the critical portion of Elk-1's TAD (308-401), the identification of a critical phosphorylated Elk-1 residue (S383), and the biochemistry to induce and validate formation of a (seemingly near-stoichiometric) MED23-Elk-1 complex (MED23Elk-1) are well thought-out and executed.
- The MED23 and MED23Elk-1 cryo-EM maps and associated models are of high quality (<3Å resolution) and the MED23Elk-1 map shows clearly identifiable density for 11 residues in the Elk-1 TAD. Interestingly, the MED23Elk-1 cryo-EM map has slightly better resolution and MED23Elk-1 particles show a considerably different orientation distribution. This is not discussed in the manuscript.
- The functional assays to validate the significance of the reported Elk-1 interaction with MED23, based on mutation of a critical residue identified through analysis of the MED23Elk-1 structure, are convincing.
- The only significant difference between the cryo-EM MED23 structure and the MED23 structure previously determined by the authors via X-ray crystallography seems to be that the N-terminus of the subunit (aa 1-50), which was stabilized by nanobody-mediated crystal contacts, is disordered in the cryo-EM map of MED23. The authors make no mention of the fact that the MED23 N-terminus is reasonably well resolved in cryo-EM maps of the entire Mediator complex, in which that portion of MED23 makes contacts with MED14 and MED15.
- The authors make the novel observations that a) Elk-1's TAD binding to MED23 requires a slight repositioning of MED23's helix H19 and, b) Elk-1 binding causes some re-organization (comparatively minor) of the MED23 structure that extends well beyond the Elk-1 binding site.
- As the authors point out, there is little structural information about interaction of transcription factors with Mediator or Mediator subunits and this study provides a detailed view of a phosphorylated, 11 amino acid portion of Elk-1 bound to MED23, largely through hydrophobic interactions.

WEAKNESSES

- Details of the MED23-Elk-1 TAD interaction are relevant to this specific complex alone. It is not clear how, or whether they are informative about Mediator-Transcription Factor (TF) interactions in general.
- As the authors indicate in the Introduction, "TFs control Mediator function, in part by recruiting Mediator to specific genomic

sequences, such as enhancers.” Their results show details of the MED23 interaction with 11 critical residues in the Elk-1 TAD transactivation domain of Elk-1. If one assumes that the Mediator mechanism is based exclusively on recruitment, and further assumes that observations about MED23 changes detected outside the context of the entire Mediator complex are informative, then the results are interesting. But those are big assumptions. If activation is only based on Mediator recruitment, the MED23Elk-1 structure does not provide any “general” conclusions. The authors seem to acknowledge the possibility that there might be more to Mediator-dependent activation than just recruitment, but present no results or substantial arguments in this regard.

- Some arguments about the possible significance of allosteric changes in the MED23 structure are presented in the Discussion, but they are completely speculative and focused mostly on Elk-1 interacting partners known to be involved in activation. The possibility that analyzing the structure of MED23Elk-1 in the context of the entire Mediator complex could lead to different conclusions is mostly ignored. As the authors point out, “How TF-Mediator interactions will impact Mediator subunits and as a result the full Mediator, remains a central question.”
- Arguments about the disease-related R617Q MED23 mutation are also completely speculative.
- The authors mention that “It is expected that TF-Mediator binding coincides with conformational changes in the Mediator complex, and this will influence Mediator function”. Unfortunately, this statement is made in the context of referencing two old (>20 years) publications that came to conclusions regarding large, long-range Mediator rearrangements induced by TF binding based on very low-resolution structures from stained particles that, now that high-resolution structures of Mediator are available, are completely artifactual. By referencing such outdated results, the authors miss the point that those old studies did not identify correct binding locations (e.g. for the CTD), and proposed large changes in Mediator organization that now are clearly incompatible with the actual structure of the complex. Arguments about TF-induced Mediator rearrangements also overlook the fact that most Mediator subunit TF targets are located in intrinsically disordered portions of Mediator subunits.

Version 1:

Reviewer comments:

Reviewer #1

(Remarks to the Author)

The authors have addressed all my concerns in the revised manuscript. I support publication of this study.

Reviewer #2

(Remarks to the Author)

All my previous concerns are addressed in the revised manuscript.

Reviewer #3

(Remarks to the Author)

After reviewing the revised manuscript and carefully considering responses to the initial round of reviewer comments, I continue to think that the manuscript by Monté et al. presents solid results that, unfortunately, do not provide insight into the mechanism of transcription regulation by Mediator. Rearrangements observed in MED23 following interaction with Elk1 are relatively minor and their mechanistic significance is impossible to assess when they take place outside the context of the entire Mediator complex. I agree with the comment from Reviewer 2 that the results presented in the manuscript do not add to our current understanding of transcriptional control mechanisms. I think that this work will have little appeal for the general audience of Nature Communications and should be published in a more specialized journal.

Responses to Reviewer comments

Reviewer #1

«Mediator is a multi-subunit complex that connects transcription factors (TFs) with the RNA Polymerase II machinery to regulate transcription. Many TFs have been reported to bind to different Mediator subunits. However, the mechanisms of TF-Mediator interactions remain unclear due to the diversity and dynamic nature of transcription factors. The work by Didier Monté et al. describes the structure of the human Mediator subunit MED23 with the activation domain of the transcription factor Elk-1. The authors show that the Elk-1 activation domain binds MED23 through a hydrophobic region, inducing conformational changes in MED23 and revealing the interaction mechanism between the MED23 subunit and Elk-1. This work provides new insights into the TF-Mediator interaction mechanism and how TFs interact with Mediator to regulate gene transcription. I support the publication of the study.»

ANSWER. This comment aptly summarizes the key message we aim to convey with this work.

«1. The multiple phosphorylation sites in Elk-1 regulate its transcriptional activation. The authors discussed that Elk-1 S383p is directly involved in MED23 binding and contributes to the stabilization of MED23-Elk-1 interaction. Would the lack of phosphorylation at S383 weaken the binding between MED23 and Elk-1?»

ANSWER. To address this question (also raised by Reviewer 2), we investigated the interaction between the Elk-1 activation domain and MED23 using pull-down assays and SPR experiments. First, we tested whether the Elk-1 activation domain could bind to MED23 when phosphorylated exclusively at S383 in the MBM peptide (Elk-1^{S383p}). We observed that Elk-1^{S383p} binds to MED23 with a K_d of 81 nM, whereas minimal interaction was detected by pull-down assays in the absence of phosphorylation. No K_d could be evaluated in this case. This is consistent with the finding that phosphorylation at least at one position in Elk-1 is required for MED23 binding and subsequent transcriptional activation.

Since Elk-1^{3P} includes two additional phosphorylation sites (T368 and S389), we further investigated their relative contributions. Details on this analysis are provided in our responses to Reviewer 2's

«2. The manuscript identifies the key residue, F378-Elk1, which plays a crucial role in the interaction between MED23 and Elk1. The authors demonstrate that the MED23-G382F mutation disrupts binding to Elk-1, thereby affecting the expression of the Elk-1 target gene Egr1. How does MED23-G382F affect the transcription levels of Elk-1 target genes in MED23-G382F mutant cells?»

ANSWER. We extended our analysis to other Elk-1 target genes, including Egr-2, Egr-3, and Fos. The mRNA levels of these target genes were quantified using real-time PCR. Consistent results were obtained. All target genes were similarly affected when the G382F mutant version of MED23 was used. These findings are discussed lines 345-347, lines 964-968 (Methods section). The abstract has been updated (lines 24-25).

«3. The authors should provide the SDS-PAGE of purified proteins to show the quality of the proteins used in the study.»

ANSWER. The SDS-PAGE of purified proteins have been provided in Figures 5b and supplementary Fig. 5b.

Reviewer #2 (Remarks to the Author):

«Monte et al. determine a cryo-EM structure of the Elk1-binding mode of human MED23 at a reported resolution of 3.0Å. They propose a model of the path of Elk1 to the binding site of MED23. As Mediator is one of the most critical co-activators for transcription, this paper potentially can capture attentions from general readers. However I feel there are several deficiencies. Most critically, it is unclear how the structural information gleaned from this study using Elk1 will specifically lead to new understandings of transcriptional mechanisms. The dissociation constant (K_d) of MED23 to this region of Elk1 should be defined. Furthermore, how much the HFWS sequence contributes to the binding in terms of K_d, as well as how phosphorylations contribute to the Elk1 binding in terms of K_d, should be elucidated. While the authors provided some information by citing previous papers, these citations are largely descriptive. Given that the authors seem to have a good expression system and pure soluble MED23, these measurements should be feasible without technical difficulties. If these binding parameters are not addressed, it is unlikely that this paper will capture the attention of general readers in Nature Communications.»

ANSWER. The impact of transcription factor–Mediator interactions is a critical question in the field. More specifically, the MED23-Elk1 interaction controls the response of many immediate early genes, which is not only important from a fundamental perspective but also holds potential for developing new therapies for pathologies in which these factors are dysregulated. In recent years, the collective work of several groups has enhanced our understanding of the eukaryotic Mediator, providing a more complete structural picture of the human complex. However, an important question remains: how do the numerous transcriptional activators and regulators interact with the different parts of Mediator?

So far, few structural details of transcription factor–Mediator interactions have been elucidated, largely due to the structural disorder and dynamics of activation domains. Here, we report the high-resolution cryo-EM structure of MED23 in complex with the multi-phosphorylated activation domain of Elk-1. To our knowledge, this is the first high-resolution complex between a transcription factor and a Mediator subunit. In this context, we believe our study will have a significant impact on the broader scientific community and represent an important step towards a better understanding of transcriptional regulation.

Following the comments of reviewer 2 on the impact of specific Elk-1 residues in the interaction with MED23, we undertook to study the interaction of the Elk-1 activation domain with MED23 by pull-down and SPR techniques (provided as a new supplementary figure 8). Mutation of the central FW motif in the MBM peptide has a drastic effect and abolishes the interaction between Elk-1 and MED23, suggesting that these hydrophobic residues constitute much of the driving force for MED23 binding. As Elk-1^{3P} includes three phosphorylation sites (T368, S383 and S389), we investigated their relative importance in MED23 binding using the SPR technique. We first tested whether the activation domain of Elk-1 is able to bind to MED23 when phosphorylated only at S383 in the MBM peptide (Elk-1^{S383P}). We observed that Elk-1^{S383P} binds to MED23 with a K_d of 81 nM, whereas only minimal interaction was detected in the absence of phosphorylation, consistent with the observation that phosphorylation at least at one position in Elk-1 is required for MED23 binding and subsequent transcriptional activation. We next assessed whether phosphorylation at T368 and S389 had an additional impact on MED23 binding. We measured K_d values of 42 nM and 60 nM for the doubly (T368^P and S383^P) and triply (Elk-1^{3P}) phosphorylated Elk-1, respectively. These values directly compare to the K_d observed for Elk-1^{S383P}, suggesting that phosphorylation at T368 and S389 have no significant additional impact on the binding of Elk-1^{S383P}. Finally, we tested whether phosphorylation at T368 and S389 were able to stimulate the binding of Elk-1 to MED23, in the absence of phosphorylation at S383. We observed a K_d of 75 nM for Elk-1^{T368P-S389P}, which compares to the K_d observed for Elk-1^{S383P}. These data suggest that phosphorylation at S383 in the MBM peptide is not a strict requirement for MED23 binding, but can be compensated by adjacent phosphorylation to the MBM peptide. It has long been suggested that phosphorylation of Elk-1 induce conformational changes within the transcription factor that are required for Mediator recruitment and further transcriptional activation. Our results are in agreement with such a model and prioritize the role of the different phosphorylation, showing that only one is located directly in the peptide of Elk-1 interacting with MED23.

These data are now reported in the results section (lines 169-189), and two additional paragraphs are also incorporated in the discussion section (lines 372-400).

« Line 76-77: “Medium to low resolutions”

It's too vague to say this. Medium to low means resolutions lower than 5-6 Å.»

ANSWER. We agree, and this expression adds nothing to the text. It has been removed

« Line 140: S383p-Elk-1 may form an internal hydrogen bond with the hydroxyl group of T381-Elk-1, as 140 well as with Q587 from MED23.

Readers cannot judge the presence of hydrogen bonding based on Fig. 1.»

ANSWER. We appreciate this remark. Given the resolution of our structure, it would be too speculative to propose a potential hydrogen bond. Therefore, we have removed this sentence.

« Line 144: No additional electron density coming from...

Cryo-EM does not image “electron density” unlike X-ray crystallography. People just say density or cryo-EM density. »

ANSWER. The word « electron » has been removed.

« Line 153: Although they do not appear to be directly involved in MED23 binding, they enhance the activation process and thus could play a role in promoting the propensity of the MBM peptide to adopt a turn conformation capable of binding to MED23. In addition, they could promote the recruitment of other protein partners.

I think this is too speculative. Authors should provide info about how much T368p and S389p as well as S383p contribute to binding of the Elk1 peptide to MED23 in terms of Kd.»

ANSWER. We agree that the sentence was too speculative. As a result, we have removed it and incorporated SPR data, which are now reported and discussed in detail. Please refer to the comment above for further information.

«Line 324: In the PIC, the concave face of MED23, which accommodates Elk-1 binding, is oriented towards the 5' end of the promoter DNA, ideally suited to interact with potential upstream DNA sequences (Extended Data Fig. 10). This statement is confusing. Is the Elk1 binding site immediately upstream of the core promoter? I thought it is several kbp away from the core promoter.»

ANSWER. The sentence has been revised to avoid confusion. It now reads (lines 485-486): 'In the PIC complex, the concave face of MED23, which accommodates Elk-1 binding, is oriented towards the 5' upstream DNA, making it ideally positioned to interact with transcription factors (now Supplementary Fig. 11).

«Table 1 The table should include map-model FSC (0.5).»

ANSWER. FSC (0.5) values have been added in Table 1.

«Extended Data Fig. 2 and Fig. 6

Micrograph and 2D classes should have scale bars.»

ANSWER. Scale bars have been added.

«Extended Figure 5b

It would help to label the bands seen to convince the reader of the purity and phosphorylation status»

ANSWER. Labeling has been added.

Reviewer #3 (Remarks to the Author):

STRENGTHS

« • The molecular analysis to identify the critical portion of Elk-1's TAD (308-401), the identification of a critical phosphorylated Elk-1 residue (S383), and the biochemistry to induce and validate formation of a (seemingly near-stoichiometric) MED23-Elk-1 complex (MED23Elk-1) are well thought-out and executed.»

COMMENT. Thanks ! It's always nice to read this kind of comment.

« • The MED23 and MED23Elk-1 cryo-EM maps and associated models are of high quality (<3Å resolution) and the MED23Elk-1 map shows clearly identifiable density for 11 residues in the Elk-1 TAD. Interestingly, the MED23Elk-1 cryo-EM map has slightly better resolution and MED23Elk-1 particles show a considerably different orientation distribution. This is not discussed in the manuscript.»

ANSWER. It is important to note that while we observed only an eleven-amino-acid peptide bound to MED23, we deposited on the grid the full Elk-1 activation domain, fused to GFP, to form the complex with MED23. The difference in the orientation distribution may stem from the additional mass, which could influence how the bound MED23 particles behave both in solution and during the vitrification process. We have added a sentence discussing this point (lines 909-912).

« • The functional assays to validate the significance of the reported Elk-1 interaction with MED23, based on mutation of a critical residue identified through analysis of the MED23Elk-1 structure, are convincing.»

COMMENT. Thanks for the approval.

« • The only significant difference between the cryo-EM MED23 structure and the MED23 structure previously determined by the authors via X-ray crystallography seems to be that the N-terminus of the subunit (aa 1-50), which was stabilized by nanobody-mediated crystal contacts, is disordered in the cryo-EM map of MED23. The authors make no mention of the fact that the MED23 N-terminus is reasonably well resolved in cryo-EM maps of the entire Mediator complex, in which that portion of MED23 makes contacts with MED14 and MED15.»

ANSWER. We agree with this point. The text has been modified to clarify that the N-terminus is involved in interactions with MED14 and MED15 in the Mediator complex (line 84).

« • The authors make the novel observations that a) Elk-1's TAD binding to MED23 requires a slight repositioning of MED23's helix H19 and, b) Elk-1 binding causes some re-organization (comparatively minor) of the MED23 structure that extends well beyond the Elk-1 binding site.»

COMMENT. This is correct

« • As the authors point out, there is little structural information about interaction of transcription factors with Mediator or Mediator subunits and this study provides a detailed view of a phosphorylated, 11 amino acid portion of Elk-1 bound to MED23, largely through hydrophobic interactions »

COMMENT. This is correct

WEAKNESSES

« • Details of the MED23-Elk-1 TAD interaction are relevant to this specific complex alone. It is not clear how, or whether they are informative about Mediator-Transcription Factor (TF) interactions in general.»

COMMENT. Transcription factors perform highly specific functions, and each factor is expected to interact with its transcriptional partners in a unique manner. However, few structural details of transcription factor-Mediator interactions are currently available, due to the structural disorder and/or dynamics of activation

domains, which hinder their study at high resolution. To the best of our knowledge, we present the first high-resolution structure of a complex between a transcription factor and a Mediator subunit. Further studies are needed to expand our understanding of Mediator–transcription factor (TF) interactions and potentially identify common features across these molecular interactions.

« • As the authors indicate in the Introduction, “TFs control Mediator function, in part by recruiting Mediator to specific genomic sequences, such as enhancers.” Their results show details of the MED23 interaction with 11 critical residues in the Elk-1 TAD transactivation domain of Elk-1. If one assumes that the Mediator mechanism is based exclusively on recruitment, and further assumes that observations about MED23 changes detected outside the context of the entire Mediator complex are informative, then the results are interesting. But those are big assumptions. If activation is only based on Mediator recruitment, the MED23Elk-1 structure does not provide any “general” conclusions. The authors seem to acknowledge the possibility that there might be more to Mediator-dependent activation than just recruitment, but present no results or substantial arguments in this regard.»

COMMENT. The impact of transcription factor recruitment by the Mediator complex and its subsequent effect on the entire PIC remains a vast and unresolved question in molecular biology. A key challenge is understanding how the diverse array of transcriptional activators and regulators interact with different parts of Mediator, through mechanisms that may involve either structured or unstructured regions, and with or without post-transcriptional modifications. Currently, there is a lack of data regarding these interactions. In this context, we provide one of the first—if not the first—high-resolution complexes between a transcription factor and a Mediator subunit.

Unlike many Mediator subunits, MED23 is structured and retains its structure when isolated from the complete Mediator complex, with very limited disordered regions. This characteristic makes it an invaluable tool for studying how transcription factors recruit the Mediator complex, and we have leveraged this tool to dissect the interaction at high resolution. While it is clear that understanding MED23's behavior within the entire Mediator complex during Elk-1 recruitment is crucial, obtaining such detailed information has not yet been feasible when studying the full Mediator.

It is worth noting that subunits like MED23, which belong to the Mediator Tail module, are not known to interact with other PIC partners (outside of Mediator), highlighting their specific role in transcription factor recruitment. However, they are also known to interact with other regulatory proteins, further emphasizing their multifaceted role in transcriptional regulation.

« • Some arguments about the possible significance of allosteric changes in the MED23 structure are presented in the Discussion, but they are completely speculative and focused mostly on Elk-1 interacting partners known to be involved in activation. The possibility that analyzing the structure of MED23Elk-1 in the context of the entire Mediator complex could lead to different conclusions is mostly ignored. As the authors point out, “How TF-Mediator interactions will impact Mediator subunits and as a result the full Mediator, remains a central question.”»

COMMENT. Thank you for this insightful comment. We agree that potential allosteric changes in the MED23 structure within the full Mediator complex will be an important aspect to explore in future studies. While our current study focuses on the MED23–Elk-1 interactions at high resolution, we recognize the need for future work to investigate how these interactions influence the Mediator complex as a whole. As we mention in the manuscript, the impact of transcription factor–Mediator interactions on the full Mediator remains a central question in the field. We hope that our work will serve as a foundation for future investigations that incorporate the entire Mediator complex, leading to a more comprehensive understanding of these interactions.

We have acknowledged this limitation in the manuscript and emphasized that further studies are needed to fully address how transcription factor–Mediator interactions impact the structure and function of the complete Mediator complex.

«• Arguments about the disease-related R617Q MED23 mutation are also completely speculative. »

ANSWER. We agree. This paragraph has been removed.

«• The authors mention that “It is expected that TF-Mediator binding coincides with conformational changes in the Mediator complex, and this will influence Mediator function”. Unfortunately, this statement is made in the context of referencing two old (>20 years) publications that came to conclusions regarding large, long-range Mediator rearrangements induced by TF binding based on very low-resolution structures from stained particles that, now that high-resolution structures of Mediator are available, are completely artifactual. By referencing such outdated results, the authors miss the point that those old studies did not identify correct binding locations (e.g. for the CTD), and proposed large changes in Mediator organization that now are clearly incompatible with the actual structure of the complex. Arguments about TF-induced Mediator rearrangements also overlook the fact that most Mediator subunit TF targets are located in intrinsically disordered portions of Mediator subunits. »

ANSWER. We agree that these old studies missed the correct assignment of the different modules of Mediator, although at the time they were inspiring. Very recently, the Asturias' lab used a well characterized nuclear receptor (NR) complex, TR, VDR and RXR α , to study how an activator may affect Mediator conformational transitions in presence of the Kinase module (CKM) (Zhao et al. Mol. Cell 2024 PMID 38955181, see also Chao et al. Mol Cell 2024 PMID 39321804). These studies revealed details of NR interaction with core Mediator and provides insight into structural changes triggered by NR binding that help explain the receptor's capacity to enable interaction of CKM-MED with MED26 and the RNA Pol II CTD. These references have been added to our manuscript (line 371; ref 54 & 55).